materials science

salt of alkylphosphinate, acrylonitrile–butadiene–styrene, flame retardant, synthesis, mechanical properties

**Author for correspondence:**
Xueqing Liu
e-mail: liuxueqing2000@163.com

This article has been edited by the Royal Society of Chemistry, including the commissioning, peer review process and editorial aspects up to the point of acceptance.

†Present address: Key Laboratory of Optoelectronic Chemical Materials and Devices (Ministry of Education), Jianghan University, Wuhan 430056, People's Republic of China.

# Synthesis of a novel aluminium salt of nitrogen-containing alkylphosphinate with high char formation to flame retard acrylonitrile–butadiene–styrene

Xue Yang[1], Hao Wang[2], Xueqing Liu[2,†] and Jiyan Liu[2,†]

[1]Institute of Noise and Vibration, Naval University of Engineering, Wuhan 430033, People's Republic of China
[2]Key Laboratory of Optoelectronic Chemical Materials and Devices (Ministry of Education), Flexible Display Materials and Technology Co-innovation Center of Hubei Province, Jianghan University, Wuhan 430056, People's Republic of China

(iD) XL, 0000-0001-6253-6386

A novel nitrogen-containing alkylphosphinate salt—aluminium β-(p-nitrobenzamide) ethyl methyl phosphinate (AlNP) was synthesized and used to flame retard acrylonitrile–butadiene–styrene copolymer (ABS). The Fourier transform infrared spectrometry, $^1$H, $^{13}$C and $^{31}$P nuclear magnetic resonance and X-ray fluorescent spectroscopy (XRF) were applied to characterize the structure and composition of products. The flame retardancy performance, thermal properties and mechanical strength of the ABS/AlNP with respect to AlNP loading were investigated. AlNP was stable before 330°C and decomposed very slowly with residues high up to 56.1% at 700°C. Adding 25–30 wt% of AlNP alone can make ABS to pass V0 rating in the vertical burning tests (UL 94). The results according to the micro combustion calorimeter, thermogravimetric analysis showed that AlNP can depress the heating release and retard the thermal degradation of the ABS. Scanning electron microscopy observation of the residues from LOI test indicated that AlNP formed the condensed and tough residues layer during combustion; XRF analysis showed that the residues contained phosphorus and aluminium element and nitrogen element was not detected. The compact phosphorus/aluminium-rich substance acted as a barrier to enhance flame-retardant properties of the ABS.

# 1. Introduction

Acrylonitrile–butadiene–styrene (ABS) is a high-performance polymer with excellent mechanical/electric properties, fine surface appearance, as well as chemical resistant. It has been widely applied in electronic and electricity industry, automobile and construction field [1–3]. ABS is highly flammable with the limited oxygen index (LOI) only approximately 18%. So, adding the flame retardant is an effective approach to reduce its flammability [4–6]. The commercial flame retardants used for ABS are halogen-containing compounds. They have been restricted in high-tech application because they release the toxic compound during burning and result in the environment and ecological problem. Phosphorus-containing compounds have been regarded as promising flame retardant to replace the halogen-containing compounds for ABS [7,8] because they are eco-friendly and effective in many polymer materials.

The efficiency of phosphorus-based flame retardants is contributed by radicals such as $PO^{\bullet}$, $PO_2^{\bullet}$ and $HPO^{\bullet}$, scavenging active radicals $H^{\bullet}$ and $OH^{\bullet}$ in the gas phase, or contributed by residual char in the condensed phase, protecting material from flammable gases, oxygen and heat. The flame-retardant performance of phosphorus compound mainly depends on phosphorus content. Recently, Braun et al. [9] and Hoang et al. [8] revealed that 'oxidation state' of the phosphorus compounds also influence flame retardancy performance of the polymer. The compounds containing high oxidation state phosphorus, for instance, organic phosphonate, produce more thermally stable residue and act mainly in the condensed phase, while the compounds containing low oxidation state phosphorus such as phosphinate and phosphonate tend to decompose into volatile gas and act in the gas phase during combustion. Hoang et al. found that organic phosphinate and phosphonate are more effective in the ABS than phosphate when the phosphorus content is in the range of 4.75–5.07 wt%. The reason is that the ABS is incapable of char forming, that condensed mode retardancy action from phosphate contributes very little to the combustion of ABS. By contrast, the gas-mode action from phosphinate and phosphonate works well in the retardancy of ABS. Generally, for the highly flammable ABS, 30 wt% of flame retardant in the formulation is required to achieve UL 94 V0 rating, and above 4.80% of phosphorus is needed to achieve self-extinguishing when a single organic phosphorus compound is added [10]. To maximize flame retardancy performance of the ABS at the lowest possible loading of flame retardant, various synergistic additives such as nitrogen- [11], silicon- [12], sulfur- [13] and boron- [14] containing compound, or inorganic nanoparticles are applied to enhance the flame retardancy of phosphorus/ABS system by quenching the active $OH^{\bullet}$ and $H^{\bullet}$ radicals, or by promoting the char formation.

The synergistic additives can combine with phosphorus compound by physical mixing or chemical connection. The physical mixing is an easy and important way. However, heterogeneous mixture will cause the deterioration of the mechanical properties of ABS [15]. By comparison, the chemical connection of multiple components in one is more stable and effective, and it endows polymer with flame retardancy, good compatibility, and the retentive physical and mechanical properties [16].

Metal salts of alkylphosphinates are now widely applied flame retardants. They showed a satisfying balance among the flame-retardant efficiency, mechanical properties and filler loading. Commercial product aluminium diethylphosphinate (DEP) is very efficient in oxygen-containing polymer such as polyamides [17], polylactide [18], polyurethane [19], poly (butylene terephthalate) [20] and epoxy [21]. However, aluminium alkylphosphinate is not efficient in the ABS below 30 wt% of loading. Realinho et al. [22] revealed that metal salts of alkylphosphinates mainly act in the gas phase and could not form the stable char in the ABS. When charring component such as ammonium polyphosphate (APP) combined with aluminium diethylphosphinate was added into ABS together, both gas and condensed mode of action can achieve and the flame retardancy of ABS was enhanced.

In this work, A novel nitrogen-containing phosphinate compound—aluminum β-(p-nitrobenzamide) ethyl methyl phosphinate (AlNP) with thermal stability above 360°C and high char formation has been synthesized from 2-methy-2, 5-dioxo-1, 2-oxaphospholane and p-nitroaniline. The AlNP was applied as flame retardant of ABS. The flame retardancy behaviour, thermal stability as well as the mechanical properties of the AlNP/ABS composites were studied.

# 2. Experimental set-up

## 2.1. Materials

ABS granules were supplied by Qimei Corp., Taiwan area). 2-methy-2, 5-dioxo-1, 2-oxaphospholane (OP, purity more than 98%) was purchased from Zhenghao company, China. P-nitroaniline, 95% ethanol,

**Figure 1.** Synthesis route for AlNP.

1,4-dioxane, isopropanol and aluminium isopropoxide were chemical reagents and provided by Guoyao Chemical reagent company, China. All above materials and reagents were used directly without further purification.

## 2.2. Procedure for synthesizing aluminium salt of β-(p-nitrobenzamide) ethyl methyl phosphinate (AlNP)

The synthesis of AlNP included two steps. First, OP reacted with p-nitroaniline by carbon amidation to produce the NP. The NP then reacted with aluminium isopropoxide by neutralization to form AlNP, as shown in figure 1.

Synthesis of NP: to a round bottom three-neck flask equipped with a magnetic stirrer, 0.5 mol (67.0 g) OP in 200 ml of 1,4-dioxane was charged. The flask was heated to 50°C and solution of 0.5 mol p-nitroaniline (69.06 g) in 100 ml of 1,4-dioxane was added by using dropping funnel within 30 min. The solution was heated to 90°C and maintained for 3 h to ensure the amidation reaction between OP and p-nitroaniline completed. The light-yellow NP was precipitated from the solution during the reaction. The solid NP was obtained as a yellow powder by filtration, followed by purification with ethanol three times, each time 100 ml ethanol was used.

Synthesis of AlNP: 68.05 g NP (0.25 mol) was dissolved in 200 ml isopropanol in a flask, and then the solution of 153.19 g (0.75 mol) $Al(OCH(CH_3)_2)_3$ in 500 ml isopropanol were charged. The neutralization reaction between NP and $Al(OCH(CH_3)_2)_3$ was carried out at 85°C for 4 h under refluxing. After the reaction completed, the flask cooled down to the ambient temperature. The yellow AlNP precipitate was filtered off and purified with ethanol. The residue solvent in the AlNP was removed with vacuum oven at 100°C for 24 h (yield = 96.7%).

## 2.3. Preparation of ABS/AlNP composites

The ABS composite with different AlNP loading was processed with a HAPRO Mix-60c mixer (Harp Electrical Technology Co., China) at 60 r.p.m. at 220°C for 10 min. Finally, the neat ABS and ABS/AlNP sheets with 3.0 mm of thickness were obtained by hot pressing at 225°C for 15 min under 10 MPa with an XLB-DQ400X plate vulcanization machine (Yadong No. 3 Rubber Co., China). The compositions of the ABS/AlNP composites are listed in table 1.

## 2.4. Measurements

The $^1H$, $^{13}C$ and $^{31}P$ nuclear magnetic resonance (NMR) spectra were performed in DMSO-$d_6$ at 25°C on a 400 MHz of Mercury VX-400 instrument (Varian, US). Tetramethyl silane was used as a reference of $^1H$, $^{13}C$ and 85% $H_3PO_4$ is used as a reference of $^{31}P$-NMR spectra. Fourier transform infrared (FTIR) measurement

**Table 1.** Thermogravimetric data for the ABS/AINP composites in $N_2$.

| sample | $T_i$ (°C) | $DTG_{max}$ (%°C$^{-1}$) | $T_{max}$ (°C) | residues at 700°C (% (cal.)) |
|---|---|---|---|---|
| AINP | 325 | 0.49 | 368 | 56.1 |
| ABS | 390 | 2.15 | 434 | 0.2 |
| ABS-AINP$_{20}$ | 354 | 1.14 | 425 | 11.5 (11.4) |
| ABS-AINP$_{25}$ | 351 | 1.08 | 420 | 14.3 (14.2) |
| ABS-AINP$_{30}$ | 349 | 1.00 | 416 | 17.2 (17.0) |

was carried out using a Tensor 27 Bruker infrared spectrometer (Bruker, German) and operated at 4 cm$^{-1}$ resolution.

The X-ray fluorescent spectroscopy (XRF) analysis of the sample was done on pressed powder pellet of 4 cm in diameter. The operation was performed on a ZSX Primus II (Rigaku, Japan) XRF spectrometer.

Thermogravimetric (TG) measurement was performed with TSDT Q600 (TA, USA) under $N_2$ atmosphere from 30°C to 700°C at heating rate of 20°C min$^{-1}$, using alumina crucibles and sample mass of approximately 9–10 mg.

Limiting oxygen index (LOI) test was conducted with a HC-2 instrument (Jiangning, China) according to ASTM D2863–77. The sheet sample was cut into 130 mm in length, 6.5 mm in width and 3.3 mm in thickness.

The size of sample bars for the vertical combustion (UL 94) test is $130 \times 13 \times 3.0$ mm$^3$. The measurements were carried out with a CZF3 instrument (Jiangning, China). A methane burner was applied. The methane was under pressure of 18 kPa at the burner inlet. The distance from the end of the bar to the burner nozzle was 25 cm. The bar was ignited two times. The burning time after the first ignition and after the second ignition is indicated as $t_1$ and $t_2$, respectively. The value of $t_1$ and $t_2$ is obtained based on the average of five specimens. The UL 94 classification was obtained according to the standard STM D3801.

Micro combustion calorimetric measurement (MCC) was performed with an MCC-2 instrument (Govmark, USA). The powder sample (approximately 7 mg) was heated from 40°C to 700°C in an aluminium crucible with heating rate 1°C s$^{-1}$. The heating atmosphere is a mixture of $N_2$ (flow rate 80 ml min$^{-1}$) and $O_2$ (flow rate 20 ml min$^{-1}$).

The glass transition temperature (Tg) was investigated with a Q-20 differential scanning calorimeter (DSC, TA, USA). The measurement was carried under $N_2$ atmosphere (flow rate 50 ml min$^{-1}$). The sample of 6–8 mg was used and heated from 30°C to 250°C at temperature rate of 10°C min$^{-1}$.

Notched Izod impact test was completed with a ZBC1400–1 tester (SANS, Shenzhen, China) following the national standard GB/T1043.1-2008. The tensile and bending strength were both performed on a CMT4503 instrument (SANS, China). The tensile test follows the national standard GB/T1040.2-2006 with 50 mm min$^{-1}$ of crosshead speed. The bending test was carried according to the national standard GB/T9341-2008 with mm$^2$ min$^{-1}$ of crosshead speed.

The morphology of carbonaceous residues of the ASB composites after complete combustion in LOI test and the fracture surfaces of specimens subjected to impact was investigated using a SU8000 scanning electronic microscope (SEM; Hitachi, Japan).

# 3. Results and discussion

## 3.1. Characterization of AINP

The synthesis of AlNP was carried out by amidation reaction and neutralization. First, OP reacted with p-nitroaniline by amidation reaction to form the NP. The amidation reaction is more difficult than neutralization, so, the structure of NP needed to be identified. Figure 2 presents the $^1$H-NMR, $^{12}$C-NMR and $^{31}$P-NMR of the NP.

As shown in the $^1$H-NMR spectrum, the shifts were observed at 1.2–1.5, 1.80 and 2.2–2.5 ppm corresponding to the protons of $-CH_3$, P-$CH_2$ and $-CH_2-C=O$. The shifts clearly observed at 4.75 and 10.1 ppm were ascribed to the protons of P–OH and N–H, respectively [23]. The double peaks at 8.13 and 7.8 ppm were from the protons of aromatic ring. There were eight signals for the carbon atoms

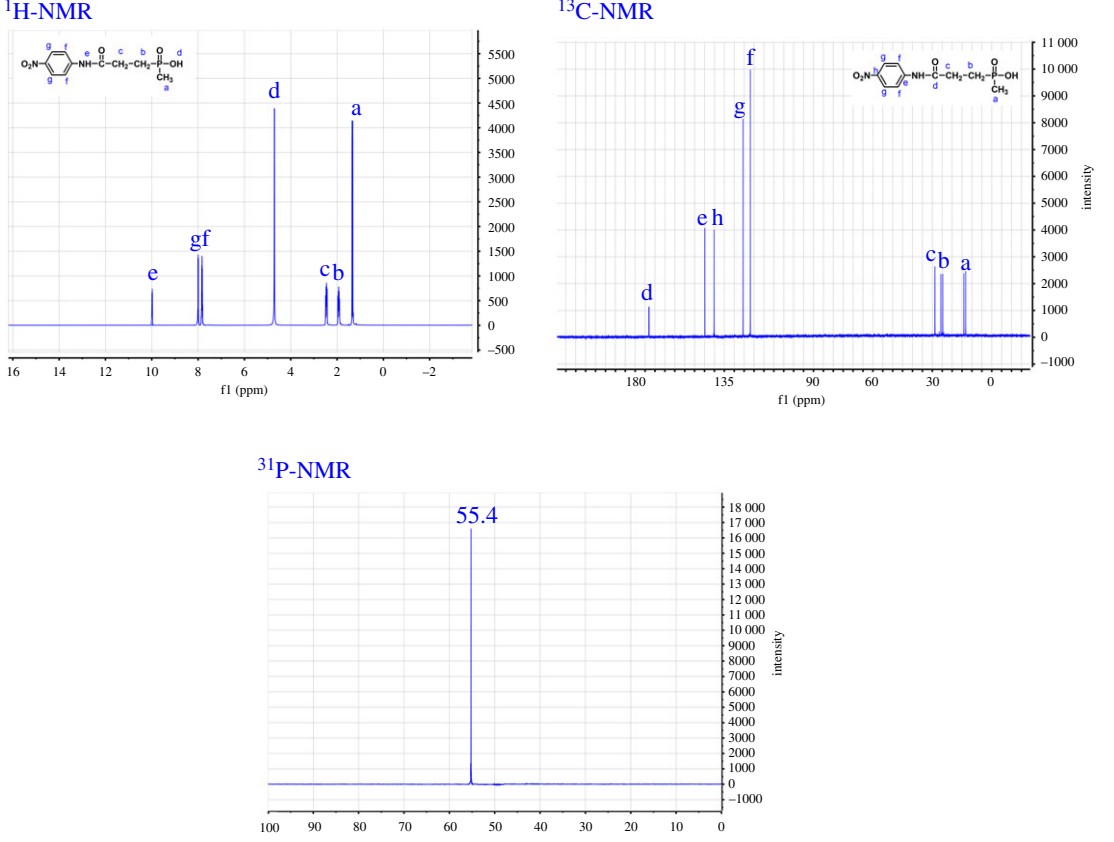

**Figure 2.** $^1$H-NMR, $^{12}$C-NMR and $^{31}$P-NMR spectra of NP.

observed in the $^{13}$C-NMR spectra of NP. The corresponding data (ppm) were summarized to the following: the shifts at 173.1, 145.3 and 142.1 were observed corresponding to carbons of C=O, C–NO$_2$ and C–NH. The signals at 129.1 and 121.8 ppm were corresponding to the carbons of ortho-position to amide group and ortho-position to the nitro group of the benzene. Signal at 28.7 ppm was for the carbon which connects to C=O. The peaks at 25.7 and 14.1 ppm were assigned to carbons of CH$_2$–P and CH$_3$–P, respectively. $^{31}$P-NMR showed that phosphorus of NP has only one chemical environment, and signal at 55.4 ppm was observed [24].

The AlNP has a similar structure to the NP except that the P–OH group of the NP has changed into the P–O–Al in the AlNP after NP reacting with aluminium isopropoxide by neutralization. AlNP was difficult to dissolve in the organic solvent, its structure was characterized with FTIR, as shown in figure 3. The characteristic absorptions at 3300, 1502, 1255 and 756 cm$^{-1}$ were for –NH group; the peaks at 3016, 2979 and 1402 cm$^{-1}$ were due to the –CH$_3$CH$_2$– group. The peak at 1700 cm$^{-1}$ was a characteristic absorption of C=O of amides. The peaks at 1598 and 690 cm$^{-1}$ were ascribed to aromatic ring; the peaks at 1550, 1338 and 850 cm$^{-1}$ were for NO$_2$ group. Other absorption peaks were: 1338 cm$^{-1}$ (P–CH$_3$), 1155 cm$^{-1}$ (P=O), 1082 and 968 cm$^{-1}$ (P–O–Al) [25]. The AlNP was amorphous, as indicated in XRD spectra. XRF result indicated that AlNP contains 3.75 wt% of Al and 12.49 wt% of P. The atomic ratio of P to Al based on the XRF was 2.90 : 1, close to calculated value of 3 : 1. The results from $^1$H-NMR, $^{13}$C-NMR, $^{31}$P-NMR, FTIR and XRF (figure 4$b$) indicated that AlNP has been synthesized successfully.

## 3.2. Thermal analysis

The thermal decomposition behaviour of AlNP and ABS/AlNP composites was characterized with TG. The relevant mass loss and derivatives of mass loss (DTG) with respect to the temperature are illustrated in figure 5. The parameters from TG results including the maximum rate of mass loss (DTG$_{max}$), temperature of DTG$_{max}$ ($T_{max}$) are listed in table 1. Both AlNP and ABS showed one-step decomposition in the nitrogen atmosphere. The AlNP started to decompose at 325°C ($T_i$, 5 wt% loss temperature) with DTG$_{max}$ only 0.49% °C$^{-1}$ at $T_{max}$ (368°C). The residues of 56.6 wt% were obtained

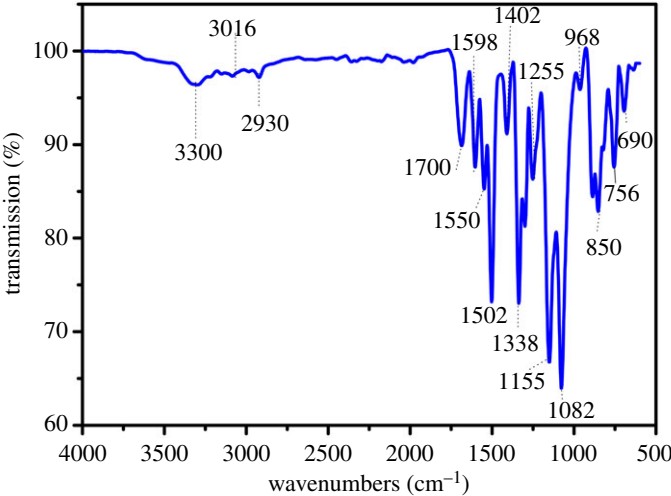

**Figure 3.** FTIR spectrum of AlNP.

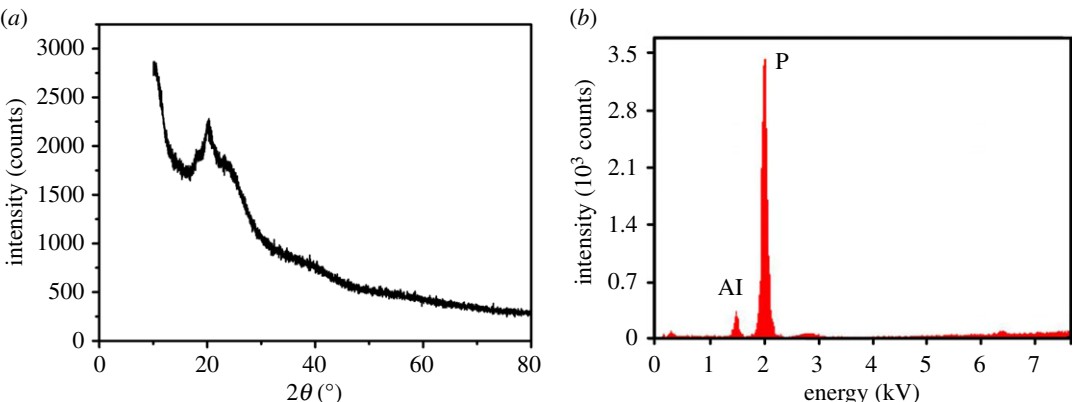

**Figure 4.** XRD (*a*) and XRF (*b*) spectra of the AlNP.

at 700°C. The ABS began to lose mass at 380°C with $DTG_{max}$ 2.15% $°C^{-1}$ at $T_{max}$ (434°C). It decomposed completely at approximately 500°C, which demonstrated that ABS has no ability of charring by itself.

All the ABS/AlNP composites exhibited two-step decomposition. The first step is minor which was related to the decomposition of AlNP. The second decomposition was mainly contributed by the ABS. It noticed that the $T_i$, $T_{max}$ and $DTG_{max}$ values for the ABS/AlNP composites shifted to a lower temperature in comparison with the pure ABS. With AlNP loading increasing, the $T_i$, $T_{max}$ and $DTG_{max}$ values decreased gradually. In the case of ABS-AlNP$_{30}$, the $T_i$ was 41°C lower and $T_{max}$ was 17°C lower than that of the pure ABS; the $DTG_{max}$ was reduced by 53.4%. This phenomenon was caused by the catalytic effect of AlNP. AlNP belongs to Lewis acid. The Lewis salts triggering ABS to decompose at low temperature have been found in the tin- and zinc-based salts/ABS system [26]. Besides, the same catalytic effect was also observed in the aluminium salt of phosphinate-filled polymer such as polyvinyl alcohol [27], EP [28] and polyamide [29].

In addition, by comparing the residues at 700°C of all the ABS/AlNP composites from the TG test with the calculated results based on the simple sum of the residues of the ABS and AlNP alone, it was found the experiment and calculated values were almost the same, indicating that the residues of ABS/AlNP were mainly contributed by the AlNP, and no extra compound was produced from chemical interaction between the AlNP and ABS. So, the AlNP suppressing the thermal decomposition of the ABS was probably due to the physical isolating of the AlNP residues.

## 3.3. Flame retardancy of ABS/AlNP composites

The effects of AlNP on the flame retardancy of ABS were evaluated with LOI value, UL 94 vertical burning test as well as the MCC measurement. All results are listed in table 2. According to the UL

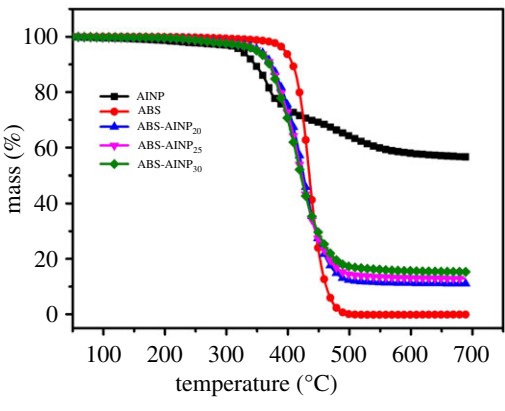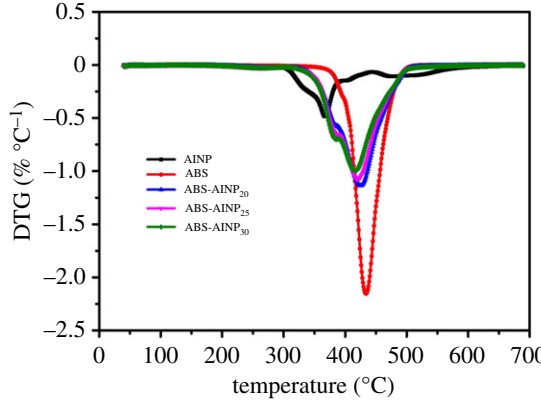

**Figure 5.** TG and DTG curves of ABS/AlNP composites in $N_2$.

**Table 2.** The LOI and vertical combustion test of the ABS composites.

| sample | PBT (wt%) | AlNP (wt%) | P (wt%) | LOI (%) | UL 94 | burning time ($t_1/t_2$) (s) | dropping | $T_P$ (°C) | PHRR (W g$^{-1}$) | THR (kJ g$^{-1}$) |
|---|---|---|---|---|---|---|---|---|---|---|
| ABS | 100 | 0 | 0 | 18.0 | NR | — | yes | 441 | 717 | 37.0 |
| ABS-AlNP$_{20}$ | 80 | 20 | 2.20 | 24.2 | V2 | 25.3/20.8 | yes | 434 | 410 | 30.8 |
| ABS-AlNP$_{25}$ | 75 | 25 | 2.75 | 26.0 | V1 | 16.6/13.2 | no | 428 | 379 | 29.6 |
| ABS-AlNP$_{30}$ | 70 | 30 | 3.30 | 27.6 | V0 | 8.4/6.4 | no | 430 | 351 | 27.0 |

94 test standard, when the afterflame time $t_1$ or $t_2$ of each individual specimen does not exceed 30 s, but the melted sample dropped and ignited the cotton pad during the test, the sample is classified as V2 rating. If the $t_1$ or $t_2$ value of individual sample does not exceed 30 s, and no dropping occurred during combustion, the sample is ranked V1 rating. When the $t_1$ or $t_2$ value of individual sample is less than 10 s with no dropping during combustion, the sample achieved V0 rating. The ABS belongs to highly flammable material. LOI value of ABS is only 18% and no rating was recorded in the UL 94 test. Adding 20 wt% AlNP can improve the LOI value of ABS to 24.2%. However, the composite only passed V2 rating. When the AlNP loading increased to 25 wt%, the LOI value of ABS rises by 1.8%; when both the $t_1$ and $t_2$ are below 10 s, the sample reaches the V0 rating. According to the data in table 2, we can speculate that 25–30 wt% of AlNP was enough for ABS to pass the V0 rating.

The dependence of heat release rate (HRR) of the ABS/AlNP composite on the temperature measured with MCC is shown in figure 6. The combustion parameters such as the maximum value of the heat release rate (PHRR) and the temperature of PHRR ($T_p$) are presented in table 2. The ABS started to release heat at approximately 360°C and HRR increased very fast after that. The HRR reached the maximum value of 717 W g$^{-1}$ at 450°C and then reduced gradually till 500°C. The total heat release (THR) was 37.0 kJ g$^{-1}$ in the temperature range of 400–500°C.

In comparison with the pure ABS, the PHRR of the ABS-AlNP$_{20}$ is reduced by 42.8% and THR was decreased by 16.7%, respectively. For the ABS-AlNP$_{30}$, the PHRR and the THR value were reduced by 51.0% and 27.2%, separately. MCC results indicated that AlNP was very efficient in depressing the heat release of the ABS combustion.

### 3.4. Residues analysis

Figures 7 and 8 show the SEM morphology of the residues of ABS and ABS/AlNP composites after the LOI test. The ABS is easy to burn; there is almost no char left after the test, as seen in figure 7. The remnant surface was wrinkled, which was caused by the shrinkage of the ABS under LOI test. For the ABS/AlNP composites (figure 8), the residues showed a discontinuous lump morphology under low magnification. At high magnification, the residues of ABS-AlNP$_{20}$ featured many spherical

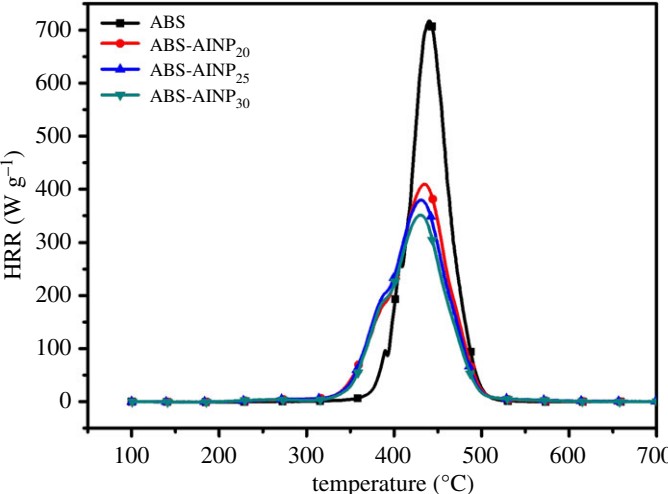

**Figure 6.** MCC curves for the ABS/AlNP composites.

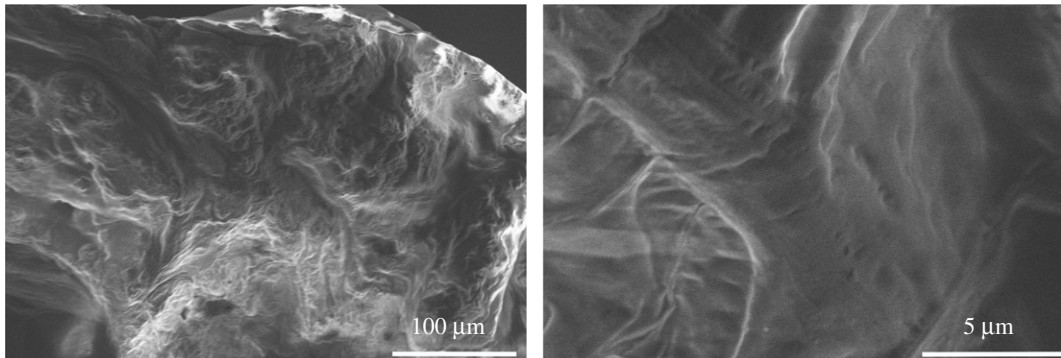

**Figure 7.** SEM morphology of residues after LOI test of ABS.

particle clusters with particle diameter in the range from 0.2 to 5 μm. The particles were isolated with a poor cohesive microstructure. When the AlNP loading increased to 25 wt% (figure 8$b$), the particles in the residues were connected cohesively (figure 8$b_1$). In the residues of ABS-AlNP$_{30}$, no round-shape particles were observed. The surface layer was condensed and smooth (figure 8$c_1$). Close observation showed that the connected residues layer was approximately 5.5 μm of thickness (figure 8$c_2$).

The spherical microparticles of different size were probably formed by the melted ABS covering the AlNP particles, followed by the charring during the combustion. It has been known that the ABS has no ability to form the char. TG results proved the ABS has released in the gas during heating. Therefore, the residues of ABS/AlNP composites mainly contributed by the AlNP. At the low AlNP loading (20 wt%), only isolated particles were left in the residues after the ABS burned into gases completely. At high AlNP loading (25 wt%), AlNP particles or particle aggregate decomposed lump residues and cohere together, so the bumpy morphology as figure 8$b_1$ was observed. When the AlNP loading increased to 30 wt%, AlNP residues completely connected and formed the smooth and continuous layer. So, adding enough AlNP was essential to prevent the ABS burning and heat spreading during combustion.

Figure 8$c_3$ presents the XRF spectra of residues of ABS-AlNP$_{30}$. The aluminium and phosphors on the surface of the residues with corresponding loading of 69.2% and 30.8% were detected. The atomic ratio of P to Al of the residues is 2.25 : 1, less than that of AlNP (2.80 : 1) from figure 3. It supposed that the aluminium was left in the residues after combustion. So, reduction in the atomic ratio of P/Al is caused by the loss of phosphorus, which has released into air. In addition, the nitrogen was not found in the residue of ABS-AlNP$_{30}$, implying it has released into gas. Based on the SEM and XRF results of the residues, it can be speculated that AlNP decomposed into phosphorus-rich residues acting as protecting layer in the condensed phase. On the other hand, AlNP released inert nitrogen-containing substance and

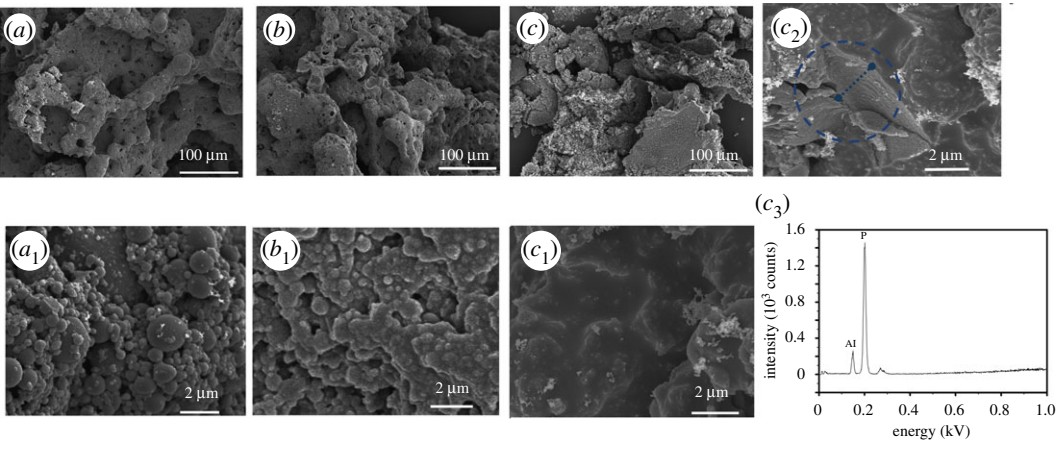

**Figure 8.** SEM morphology of residues after LOI test of ABS/AlNP composites: $(a, a_1)$ ABS-AlNP$_{20}$; $(b, b_1)$ ABS-AlNP$_{25}$; $(c, c_2)$ ABS-AlNP$_{30}$; $(c_3)$ XRF spectrum of residues from ABS-AlNP$_{30}$.

**Table 3.** Mechanical properties of ABS/AlNP composites.

| sample | tensile strength (MPa) | Young's modulus (GPa) | elongation at breaking (%) | bending strength (MPa) | impact strength (kJ m$^{-2}$) |
|---|---|---|---|---|---|
| ABS | 44.3 ± 2.0 | 2.24 ± 0.05 | 16.0 ± 1.2 | 57.8 ± 1.8 | 15.3 ± 1.0 |
| ABS-AlNP$_{20}$ | 39.7 ± 1.9 | 2.40 ± 0.05 | 14.5 ± 0.3 | 51.4 ± 2.1 | 13.2 ± 0.9 |
| ABS-AlNP$_{25}$ | 37.4 ± 1.5 | 2.48 ± 0.11 | 13.4 ± 0.4 | 48.8 ± 2.0 | 12.8 ± 0.7 |
| ABS-AlNP$_{30}$ | 35.5 ± 1.6 | 2.55 ± 0.08 | 12.3 ± 0.2 | 46.9 ± 1.9 | 11.02 ± 0.6 |

phosphorus active radicals, diluting and quenching the flammable active radicals in gas phase. This conclusion was consistent with our assumption and agrees with the results from TG and MCC analysis.

## 3.5. Mechanical properties and glass transition temperature of ABS/AlNP composites

The mechanical properties including tensile strength, Young's modulus, elongation at breaking, bending and impact strength of the AlNP/ABS composites were studied, the results are shown in table 3. Adding AlNP promoted the modulus of the ABS, but the tensile, bending and impact strength of the ABS were deteriorated. For instance, the modulus for ABS was 2.24 GPa, it increased to 2.40 GPa for ABS-AlNP$_{20}$ and reached 2.55 GPa for the ABS-AlNP$_{30}$. When 30 wt% of the AlNP loading was added, the tensile strength, bending strength and impact strength of the ABS/AlNP composites were reduced by 19.9%, 18.9% and 26.7%, respectively. The elongation of the composite was reduced by 16.3%.

For the composites obtained by physical mixing of multiple components, modulus of the composite was a sum of the modulus fraction of the individual component. The AlNP was more rigid than ABS, the ABS/AlNP composite showed a higher modulus than the pure ABS. The strength of the ABS/AlNP composites relies on the dispersion of the AlNP and interface adhesion between the AlNP particles and the ABS resin. With smaller particle size and stronger interface adhesion, the stress transferred efficiently between matrix and particles, and the composites showed an enhanced mechanical property.

In order to investigate the dispersion of the AlNP in the ABS, the fracture surface of the ABS and ABS/AlNP composites after impacting test were subjected to SEM observation, as shown in figure 9. Pure ABS exhibited a typical brittle failure feature. Compared to the pure ABS, the surface of the ABS/AlNP is rougher, indicating the ABS/AlNP composites are more brittle than the ABS. In the surface of the fractured ABS/AlNP composites, the white AlNP particles and particle aggregates with the size from 0.2 to 2 μm distributed uniformly within the ABS. In addition, there were caves on the surface. The caves became larger with AlNP loading increasing. These caves were produced by the

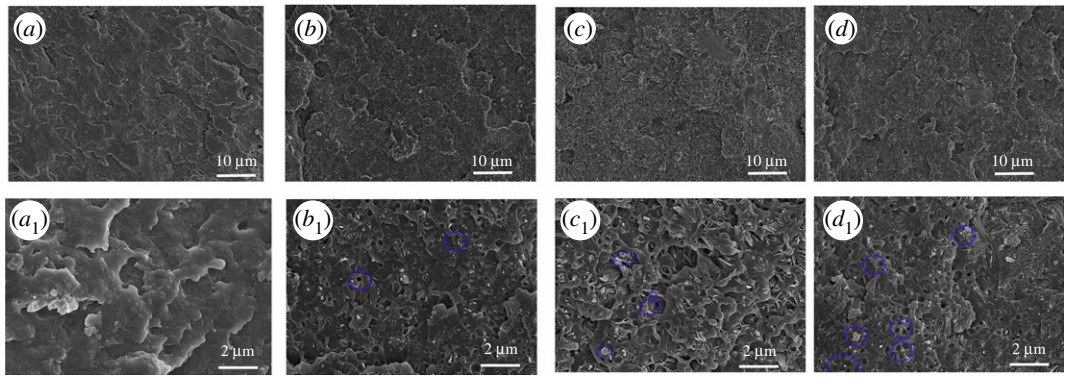

**Figure 9.** SEM morphology of fracture surfaces of the ABS/AlNP composites: (a,a$_1$) ABS; (b,b$_1$) ABS-AlNP$_{20}$; (c,c$_1$) ABS-AlNP$_{25}$; (d,d$_1$) ABS-AlNP$_{30}$.

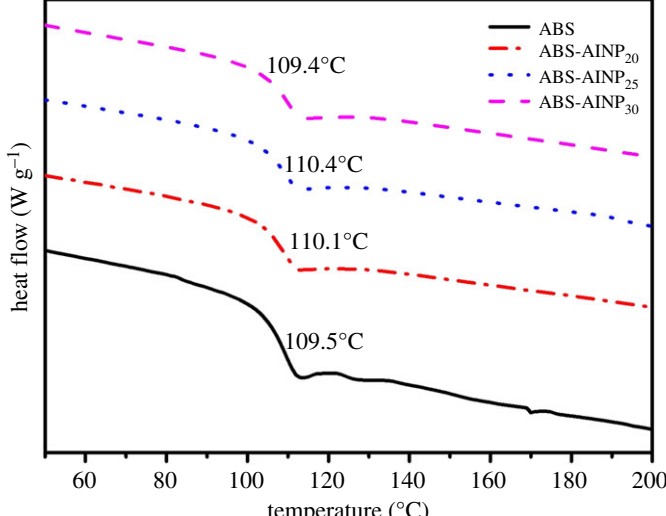

**Figure 10.** DSC curves of ABS/AlNP composites.

AlNP aggregate pulling out of the ABS matrix in the impacting test. These large aggregates could not transfer the stress to the matrix as well as the small AlNP particle, resulting in microscopic cracks of the composite. From the SEM observation, the poor dispersion of the AlNP and weak interface strength were responsible for the deterioration the ABS/AlNP composites.

It was known that the alkyl group of the phosphinate related to the polarity and hydrophobicity of particle surface. Good match in the polarity and hydrophobicity between the particle surface and polymer matrix will improve the interface strength and enhance the mechanical properties of the final composites. According to the report from the literature [30], the impact strength of ABS containing 28 wt% DEP was reduced by 50.2% compared to the pure ABS. For the ASB containing 30 wt% of (ABS-AlNP$_{30}$), the impact strength was reduced by 26.7%. That is, AlNP has a less negative effect on the mechanical properties of ABS and shows a better dispersibility in ABS matrix than the DEP.

The glass transition temperature (Tg) obtained by DSC curves were presented in figure 10. Adding AlNP did not influence the Tg of the ABS. ABS/AlNP shows the similar Tg value as the pure ABS. In addition, the Tg values of ABS/AlNP with different AlNP loading were close each other. It is approximately 110°C. Tg was related to the movement of the polymer chain segment. AlNP particle was the inert filler for the ABS, the ABS molecular chain was just absorbed on the AlNP particle surface physically. The interaction between particle and matrix was weak and the movement of polymer chain was not affected by the presence of the AlNP. So, the Tg value of ABS/AlNP was close to that of the pure ABS.

# 4. Conclusion

The AlNP was synthesized successfully and confirmed with FTIR, [1]H, [13]C and [31]P-NMR and XRF. AlNP was thermal stable and retained 56.1% of char residues at 700°C. Adding 25–30 wt% of AlNP alone made the ABS to pass UL 94 V0 test. TG analysis showed that AlNP triggered ABS to start decomposition at a lower temperature by catalyst effect but depressed the thermal degradation of ABS at higher temperature. SEM morphology revealed that AlNP can form the condensed and cohesive residues which impede the heat transfer and combustible gas spread. Subsequently, the flame retardancy of the ABS was promoted. XRF analysis indicated that the residues were rich in phosphorus and aluminium but no nitrogen element was found. The AlNP acted primarily in the solid phase. Simultaneously, AlNP released inert nitrogen-containing substance and active phosphorus-containing radicals to dilute and quench flammable substance in the gas phase.

Data accessibility. This article has no additional data.

Authors' contributions. X.Y. drafted the article; H.W. took charge acquisition of data; J.L. contributed to conception and design; X.L. finally approved of the version to be submitted.

Competing interests. We have no competing interests.

Funding. X.Y.: National Natural Science Foundation of China (grant no. 51303209). H.W. and J.L.: National Key Research and Development Program of China (grant no. 2016YFB0401505). X.L.: Key Scientific and Technological Project of Wuhan City (grant no. 2018010401011279).

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
