## [Reviewer comments · Royal Society Open Science]

Review History

RSOS-200800.R0 (Original submission)

Review form: Reviewer 1

Is the manuscript scientifically sound in its present form?

No

Are the interpretations and conclusions justified by the results?

Yes

Is the language acceptable?

Yes

Do you have any ethical concerns with this paper?

No

Have you any concerns about statistical analyses in this paper?

No

Recommendation?

Major revision is needed (please make suggestions in comments)

Comments to the Author(s)

Comments to Authors:

Ref. No.: RSOS-200800

Title: Synthesis of a novel aluminum salt of nitrogen-containing alkylphosphinate with high char formation to flame retard acrylonitrile-butadiene-styrene

Overview and general recommendation:

In this study, AINP has been successfully synthesized and characterized by the authors. Then it was used as a novel flame retardant for ABS. The ABS/AINP composites were systematically studied for their thermal stability, flame retardancy, and mechanical properties. The detailed mechanisms were also studied. Generally, the authors proposed a new flame retardant for ABS and gave us insight into how this new flame retardant can improve the flame retardancy and affect the mechanical properties of ABS. However, several sentences in the manuscript are expressed unclearly and a deeper analysis of the mechanism is expected to gain from this study.

Major comments:

1. To study the flame retardancy of polymers, the more common way is to study the thermal stability of polymers first, because it can help to understand the combustion process and flame retardant mechanism of polymers. However, the authors of this manuscript did this in a different way. When the authors discussed the flame retardancy of ABS in 3.2 Flame retardancy of ABS/AINP composites, there lacks some fundamental analysis. It might be better if thermal stability and flame retardancy can be discussed together with 3.3 Char residues analysis.
2. In Line 9-10 of Page 3, please check this sentence "The reason for that is the char layer from the phosphate is too weak to act efficiently in the condensed phase, while the phosphinate and phosphonate can work well in the gas phase". The meaning of this sentence is unclear. So will phosphate be effective in the gas phase?
3. In Line 60 of Page 4, for "A Bunsen burner with heat flux of 35 kW m⁻²", please check this sentence. In UL 94, there is no requirement of heat flux.
4. Before ABS and AINP being mixed in the HAPRO Mix-60c mixer, is there any procedure like being pre-dried for a certain time? This procedure can ensure no moisture is contained in the composite.
5. In 3.1 Characterization of AINP, when the authors did the NMR and FTIR analysis, what database did you refer to? Please include this in the manuscript.
6. In 3.3 Char residues analysis, why did the authors use the char residues after the LOI test? After UL 94 and MCC test, is there any char left? Is there any difference between them?
7. In Line 31-32 of Page 10, given the low content of nitrogen in AINP, the dilution effect of the nitrogen-containing substance is very limited to the flammable active radicals in the gas phase.
8. In Line 8 of Page 12, the authors mentioned that no interaction occurred between the AINP and ABS, based on the comparison of the residues at 700 °C of all the ABS/AINP composites from the TG test. However, the authors also mentioned the catalytic effect of AINP on ABS. These two statements are contrary to each other.
9. The almost unchanged glass transition temperature and the deteriorated strength of ABS/AINP composites both demonstrate the poor compatibility of AINP with ABS. In the second paragraph of 3.5 Mechanical properties and Glass transition temperature of ABS/AINP, the statement there is not consistent with this fact.
10. In the conclusions, the authors stated that AINP is a satisfying flame-retardant candidate for the high flammable ABS. Given the high loading of AINP, undesired compatibility with ABS, and deteriorated strength of ABS/AINP composites, this statement is misleading.
11. There are many grammar and typo errors in this manuscript. It is expected that the authors can correct them accordingly.

It should be noted the page number is for the page number shown in the manuscript for review, in which Author-supplied statements are the first page.

Review form: Reviewer 2

Is the manuscript scientifically sound in its present form?

No

Are the interpretations and conclusions justified by the results?

Yes

Is the language acceptable?

Yes

Do you have any ethical concerns with this paper?

No

Have you any concerns about statistical analyses in this paper?

No

Recommendation?

Accept with minor revision (please list in comments)

Comments to the Author(s)

In this paper, a novel aluminum salt of nitrogencontaining alkylphosphinate was designed and synthesized, and its structure was characterized in detail. AINP showed good flame retardant effect for ABS. The research topic is important and results are interesting. My general comment is that the manuscript should be accepted after minor comments.

1. The residues of ABS after burning should be given.
2. The char residues should be discussed in detail in order to explain the good flame retardant effect.
3. The SEM morphology of fracture surfaces of ABS should be given.

Decision letter (RSOS-200800.R0)

Dear Dr Liu:

Title: Synthesis of a novel aluminum salt of nitrogen-containing alkylphosphinate with high char formation to flame retard acrylonitrile -butadiene-styrene
Manuscript ID: RSOS-200800

The editor assigned to your manuscript has now received comments from reviewers. We would like you to revise your paper in accordance with the referee and Subject Editor suggestions which

can be found below (not including confidential reports to the Editor). Please note this decision does not guarantee eventual acceptance.

Please submit your revised paper before 25-Jul-2020. Please note that the revision deadline will expire at 00.00am on this date. If we do not hear from you within this time then it will be assumed that the paper has been withdrawn. In exceptional circumstances, extensions may be possible if agreed with the Editorial Office in advance. We do not allow multiple rounds of revision so we urge you to make every effort to fully address all of the comments at this stage. If deemed necessary by the Editors, your manuscript will be sent back to one or more of the original reviewers for assessment. If the original reviewers are not available we may invite new reviewers.

On behalf of the Subject Editor Professor Anthony Stace and the Associate Editor Dr Chaohua Cui.

RSC Associate Editor:
Comments to the Author:
(There are no comments.)

RSC Subject Editor:
Comments to the Author:
(There are no comments.)

Reviewers' Comments to Author:

Reviewer: 1

Comments to the Author(s)

Comments to Authors:

Ref. No.: RSOS-200800

Title: Synthesis of a novel aluminum salt of nitrogen-containing alkylphosphinate with high char formation to flame retard acrylonitrile-butadiene-styrene

Overview and general recommendation:

In this study, AINP has been successfully synthesized and characterized by the authors. Then it was used as a novel flame retardant for ABS. The ABS/AINP composites were systematically studied for their thermal stability, flame retardancy, and mechanical properties. The detailed mechanisms were also studied. Generally, the authors proposed a new flame retardant for ABS and gave us insight into how this new flame retardant can improve the flame retardancy and affect the mechanical properties of ABS. However, several sentences in the manuscript are expressed unclearly and a deeper analysis of the mechanism is expected to gain from this study.

Major comments:

1. To study the flame retardancy of polymers, the more common way is to study the thermal stability of polymers first, because it can help to understand the combustion process and flame retardant mechanism of polymers. However, the authors of this manuscript did this in a different way. When the authors discussed the flame retardancy of ABS in 3.2 Flame retardancy of ABS/AINP composites, there lacks some fundamental analysis. It might be better if thermal stability and flame retardancy can be discussed together with 3.3 Char residues analysis.
2. In Line 9-10 of Page 3, please check this sentence "The reason for that is the char layer from the phosphate is too weak to act efficiently in the condensed phase, while the phosphinate and phosphonate can work well in the gas phase". The meaning of this sentence is unclear. So will phosphate be effective in the gas phase?
3. In Line 60 of Page 4, for "A Bunsen burner with heat flux of 35 kW m⁻²", please check this sentence. In UL 94, there is no requirement of heat flux.
4. Before ABS and AINP being mixed in the HAPRO Mix-60c mixer, is there any procedure like being pre-dried for a certain time? This procedure can ensure no moisture is contained in the composite.
5. In 3.1 Characterization of AINP, when the authors did the NMR and FTIR analysis, what database did you refer to? Please include this in the manuscript.
6. In 3.3 Char residues analysis, why did the authors use the char residues after the LOI test? After UL 94 and MCC test, is there any char left? Is there any difference between them?
7. In Line 31-32 of Page 10, given the low content of nitrogen in AINP, the dilution effect of the nitrogen-containing substance is very limited to the flammable active radicals in the gas phase.
8. In Line 8 of Page 12, the authors mentioned that no interaction occurred between the AINP and ABS, based on the comparison of the residues at 700 °C of all the ABS/AINP composites from the TG test. However, the authors also mentioned the catalytic effect of AINP on ABS. These two statements are contrary to each other.
9. The almost unchanged glass transition temperature and the deteriorated strength of ABS/AINP composites both demonstrate the poor compatibility of AINP with ABS. In the second paragraph of 3.5 Mechanical properties and Glass transition temperature of ABS/AINP, the statement there is not consistent with this fact.
10. In the conclusions, the authors stated that AINP is a satisfying flame-retardant candidate for the high flammable ABS. Given the high loading of AINP, undesired compatibility with ABS, and deteriorated strength of ABS/AINP composites, this statement is misleading.
11. There are many grammar and typo errors in this manuscript. It is expected that the authors can correct them accordingly.

It should be noted the page number is for the page number shown in the manuscript for review, in which Author-supplied statements are the first page.

Reviewer: 2

Comments to the Author(s)

In this paper, a novel aluminum salt of nitrogen-containing alkylphosphinate was designed and synthesized, and its structure was characterized in detail. AINP showed good flame retardant effect for ABS. The research topic is important and results are interesting. My general comment is that the manuscript should be accepted after minor comments.

1. The residues of ABS after burning should be given.
2. The char residues should be discussed in detail in order to explain the good flame retardant effect.
3. The SEM morphology of fracture surfaces of ABS should be given.

Author's Response to Decision Letter for (RSOS-200800.R0)

See Appendix A.

RSOS-200800.R1 (Revision)

Review form: Reviewer 1

Is the manuscript scientifically sound in its present form?

Yes

Are the interpretations and conclusions justified by the results?

Yes

Is the language acceptable?

Yes

Do you have any ethical concerns with this paper?

No

Have you any concerns about statistical analyses in this paper?

No

Recommendation?

Accept with minor revision (please list in comments)

Comments to the Author(s)

Comments to Authors:

Ref. No.: RSOS-200800

Title: Synthesis of a novel aluminum salt of nitrogen-containing alkylphosphinate with high char formation to flame retard acrylonitrile-butadiene-styrene

Overview and general recommendation:

Compared with the previous version, this manuscript has been improved a lot. It is more scientifically sound. However, there are still some minor errors in the manuscript. Before accepting for publication, it is expected that the authors can correct them thoroughly.

Major comments:

1. For the flame retardancy of polymers, char usually is made of carbon, which is produced from the consumption of polymeric materials. However, in this manuscript, the residues after the decomposition of ABS/AINP composites are mainly from AINP, which includes a very limited amount of carbon. The authors also mentioned this point in the section of 3.2 Thermal analysis. Therefore, the protective layer formed by AINP is more like a ceramic-like structure, rather than a char layer. The authors should correct this point in the manuscript.
2. "The AINP contributed to flame inhibition mainly by compact phosphorus/aluminium-rich residues in condensed phase". For this statement in the Abstract, it is incorrect. Flame inhibition means the reaction in the gas phase.
3. In Line 19 of Page 21, no Figure 8b2 was included in this manuscript. Please put the right figure number here.
4. There are still many typo errors in the manuscript. Just name a few of them here: "retadancy" in Line 30 of Page 12; "STM D3801" in Line 1 of Page 15...Please do careful proofreading and check the grammar of the whole manuscript before the submission of the manuscript.

Decision letter (RSOS-200800.R1)

Dear Dr Liu:

Title: Synthesis of a novel aluminum salt of nitrogen-containing alkylphosphinate with high char formation to flame retard acrylonitrile-butadiene-styrene
Manuscript ID: RSOS-200800.R1

Thank you for submitting the above manuscript to Royal Society Open Science. On behalf of the Editors and the Royal Society of Chemistry, I am pleased to inform you that your manuscript will be accepted for publication in Royal Society Open Science subject to minor revision in accordance with the referee suggestions. Please find the reviewers' comments at the end of this email.

The reviewers and handling editors have recommended publication, but also suggest some minor revisions to your manuscript. Therefore, I invite you to respond to the comments and revise your manuscript.

Because the schedule for publication is very tight, it is a condition of publication that you submit the revised version of your manuscript before 01-Aug-2020. Please note that the revision deadline will expire at 00.00am on this date. If you do not think you will be able to meet this date please let me know immediately.

Kind regards,
Dr Laura Smith
Publishing Editor, Journals

On behalf of the Subject Editor Professor Anthony Stace and the Associate Editor Dr Chaohua Cui.

RSC Associate Editor:
Comments to the Author:
(There are no comments.)

RSC Subject Editor:
Comments to the Author:

(There are no comments.)

Reviewer comments to Author:

Reviewer: 1

Comments to the Author(s)

Comments to Authors:

Ref. No.: RSOS-200800

Title: Synthesis of a novel aluminum salt of nitrogen-containing alkylphosphinate with high char formation to flame retard acrylonitrile -butadiene-styrene

Overview and general recommendation:

Compared with the previous version, this manuscript has been improved a lot. It is more scientifically sound. However, there are still some minor errors in the manuscript. Before accepting for publication, it is expected that the authors can correct them thoroughly.

Major comments:

1. For the flame retardancy of polymers, char usually is made of carbon, which is produced from the consumption of polymeric materials. However, in this manuscript, the residues after the decomposition of ABS/AlNP composites are mainly from AlNP, which includes a very limited amount of carbon. The authors also mentioned this point in the section of 3.2 Thermal analysis. Therefore, the protective layer formed by AlNP is more like a ceramic-like structure, rather than a char layer. The authors should correct this point in the manuscript.
2. "The AlNP contributed to flame inhibition mainly by compact phosphorus/aluminium-rich residues in condensed phase". For this statement in the Abstract, it is incorrect. Flame inhibition means the reaction in the gas phase.
3. In Line 19 of Page 21, no Figure 8b2 was included in this manuscript. Please put the right figure number here.
4. There are still many typo errors in the manuscript. Just name a few of them here: "retadancy" in Line 30 of Page 12; "STM D3801" in Line 1 of Page 15...Please do careful proofreading and check the grammar of the whole manuscript before the submission of the manuscript.

Author's Response to Decision Letter for (RSOS-200800.R1)

See Appendix B.

Decision letter (RSOS-200800.R2)

Dear Dr Liu:

Title: Synthesis of a novel aluminum salt of nitrogen-containing alkylphosphinate with high char formation to flame retard acrylonitrile -butadiene-styrene
Manuscript ID: RSOS-200800.R2

It is a pleasure to accept your manuscript in its current form for publication in Royal Society Open Science. The chemistry content of Royal Society Open Science is published in collaboration with the Royal Society of Chemistry.

On behalf of the Subject Editor Professor Anthony Stace and the Associate Editor Dr Chaohua Cui.

RSC Associate Editor
Comments to the Author:
(There are no comments.)

Reviewer(s)' Comments to Author:

Appendix A

Dear Dr Laura Smith,

We thank both reviewers and editor for their positive and constructive comments and suggestions. We benefit a lot from these comments. It not only helps us to improve the quality of the manuscript, but also enhances our research ability in the future study.

We have finished the amendment of manuscript (No.RSOS-200800) entitled '**Synthesis of a novel aluminium salt of nitrogen-containing alkylphosphinate with high char formation to flame retard acrylonitrile -butadiene-styrene**', according to the comments and suggestions of reviewers, and responded, point by point to the comments. The paper has been revised throughout the text. The amendments have been highlighted with blue in the revised manuscript.

In addition, we upload all the figures and tables in a separate sheet because we worried the resolution of the Figures in the manuscript is not high enough

We would like to resubmit it to '**RSOS**', and hope it is acceptable for publication in the journal.

Sincerely

Dr. Xueqing LIU

Reply to Reviewer 1

Reviewer: 1

Comments to the Author(s)

Comments to Authors:

Ref. No.: RSOS-200800

Title: Synthesis of a novel aluminum salt of nitrogen-containing alkylphosphinate with high char formation to flame retard acrylonitrile –butadiene–styrene

Overview and general recommendation:

In this study, AINP has been successfully synthesized and characterized by the authors. Then it was used as a novel flame retardant for ABS. The ABS/AINP composites were systematically studied for their thermal stability, flame retardancy, and mechanical properties. The detailed mechanisms were also studied. Generally, the authors proposed a new flame retardant for ABS and gave us insight into how this new flame retardant can improve the flame retardancy and affect the mechanical properties of ABS. However, several sentences in the manuscript are expressed unclearly and a deeper analysis of the mechanism is expected to gain from this study.

Major comments:

1. To study the flame retardancy of polymers, the more common way is to study the thermal stability of polymers first, because it can help to understand the combustion

process and flame retardant mechanism of polymers. However, the authors of this manuscript did this in a different way. When the authors discussed the flame retardancy of ABS in 3.2 Flame retardancy of ABS/AINP composites, there lacks some fundamental analysis. It might be better if thermal stability and flame retardancy can be discussed together with 3.3 Char residues analysis.

Reply: We have adjusted the order; the thermal stability is listed as section 3.2, and the thermal stability and flame retardancy have been discussed together. Please see the revised manuscript.

2. In Line 9-10 of Page 3, please check this sentence “The reason for that is the char layer from the phosphate is too weak to act efficiently in the condensed phase, while the phosphinate and phosphonate can work well in the gas phase”. The meaning of this sentence is unclear. So, will phosphate be effective in the gas phase?

Reply: Have corrected. The revised sentence is as following: The reason is that the ABS is incapable of forming the char, that condensed mode retardancy action from phosphate contributed very little to combustion of ABS, while the gas-mode action from phosphinate and phosphonate works well in the retardancy of ABS.

Phosphate is not effective as good as phosphinate or phosphonate in the gas phase, but It is better in the condensed phase.

3. In Line 60 of Page 4, for “A Bunsen burner with heat flux of 35 kW m⁻²”, please check this sentence. In UL 94, there is no requirement of heat flux.

Reply: Corrected accordingly. It should be 'A methane burner was applied. The methane was under pressure of 18 KPa at the burner inlet.

4. Before ABS and AINP being mixed in the HAPRO Mix-60c mixer, is there any procedure like being pre-dried for a certain time? This procedure can ensure no moisture is contained in the composite.

Reply: Both ABS and AINP were dried at 100 for 8 hours in the vacuum oven before mixing.

5. In 3.1 Characterization of AINP, when the authors did the NMR and FTIR analysis, what database did you refer to? Please include this in the manuscript.

Reply: Done correctly. The references for NMR and FTIR analysis of AINP were listed as [24], [25], and [26] in the manuscript.

24. I. L. Odinets, E. A. Antonov, P. V. Petrovsky, Yu. I. Morozik, Yu. A. Krivolapov, B. I. Freger, V. Ya. Starkov, T. A. Mastryukova, M. I. Kabachnik. 1993. Interaction of 2-methyl - 2,5- dioxo-1,2-oxaphospholane with trimethylsilyl cyanide, Russian Chemical Bulletin, 42(1), 154-161.
25. N. N. Godovikov, N. A. Kardanov, T. A. Chepaikina, M. I. Kabachnik. 1982. Synthesis of some n-substituted β -aminoethyl esters of α -hydroxyphosphinic acids and their methiodides. 30(12), 2355-2359. Chemischer Informations dienst.
26. H. Krawczyk. 1996. A simple synthesis of 2-substituted 2-aminoethylphenylphosphinic acids. Phosphorus Sulfur and Silicon and the Related Elements, 118(1):195-204.

6. In 3.3 Char residues analysis, why did the authors use the char residues after the LOI test? After UL 94 and MCC test, is there any char left? Is there any difference between them?

Reply: Because the LOI test was conducted by adjusting the oxygen concentration and the sample was ignited on the top. It is easy and convenient to obtain the enough char

residues for further SEM examination in our study.

For the UL 94 test, the sample is ignited on the bottom end. When the AINP loading is low (20wt%), the sample melted and dropped onto the cotton pad and ignited the cotton. It is difficult to collect the residues. The sample for the MCC test is only about 20 mg, the residues are too little to do the further test.

We thought there is different in the SEM morphologies for the residues obtained by LOI, UL 94 and MCC test.

In the literature, morphology characterization for the residues can be carried with either LOI or UL 94 test. There was report on the characterization of the residues was from the Cone calorimeter test.

7. In Line 31-32 of Page 10, given the low content of nitrogen in AINP, the dilution effect of the nitrogen-containing substance is very limited to the flammable active radicals in the gas phase.

Reply: Have corrected accordingly. The dilution effect results from the nitrogen compounds and quenching the active radicals by phosphorus compounds worked together in the gas. So, we revised this sentence as following: 'AINP released inert nitrogen-containing substance and phosphorus-containing radicals, diluting and quenching the flammable active radicals in gas phase.'

8. In Line 8 of Page 12, the authors mentioned that no interaction occurred between the AINP and ABS, based on the comparison of the residues at 700 °C of all the ABS/AINP

composites from the TG test. However, the authors also mentioned the catalytic effect of AINP on ABS. These two statements are contrary to each other.

Reply: We did not indicate clearly here. In the early thermal decomposition stage, the AINP can catalyze the decomposition of the ABS. With the temperature increasing, the AINP decomposed and formed the condensed char to retard the decomposition of the ABS. It should be 'no extra compounds produced from chemical interaction between the AINP and ABS'. We have made correction in the manuscript.

9. The almost unchanged glass transition temperature and the deteriorated strength of ABS/AINP composites both demonstrate the poor compatibility of AINP with ABS. In the second paragraph of 3.5 Mechanical properties and Glass transition temperature of ABS/AINP, the statement there is not consistent with this fact.

Reply: Corrected accordingly. We have adjusted this paragraph. In this section, the SEM morphologies were used to explain why glass transition temperature of the ABS/AINP composites unchanged and the strength of ABS/AINP composites deteriorated. And then, we compared the mechanical properties of the ABS/AINP with ABS/DEP to illustrate the effect of the polarity of the alkyl group of the aluminum salt of phosphinate on compatibility between the filler and ABS matrix, result showed that the AINP shows better dispersibility in the ABS than the commercial product diethylphosphinate(DEP).

10. In the conclusions, the authors stated that AINP is a satisfying flame-retardant candidate for the highly flammable ABS. Given the high loading of AINP, undesired

compatibility with ABS, and deteriorated strength of ABS/AINP composites, this statement is misleading.

Reply: We have revised the conclusion sections and deleted the inappropriate states.

11. There are many grammar and typo errors in this manuscript. It is expected that the authors can correct them accordingly.

Reply: We have checked the whole manuscript and amended the error.

It should be noted the page number is for the page number shown in the manuscript for review, in which Author-supplied statements are the first page.

Reply to Reviewer: 2

Comments to the Author(s)

In this paper, a novel aluminum salt of nitrogencontaining alkylphosphinate was designed and synthesized, and its structure was characterized in detail. AINP showed good flame retardant effect for ABS. The research topic is important, and results are interesting. My general comment is that the manuscript should be accepted after minor comments.

1. The residues of ABS after burning should be given.

Reply: added the residues of ABS after burning. please see the revised Figure 7.

Figure 7. SEM morphology of residues after LOI test of ABS.

2. The char residues should be discussed in detail in order to explain the good flame retardant effect.

Reply: Have corrected accordingly. Please the revised section 3.4 Char residues analysis.

3. The SEM morphology of fracture surfaces of ABS should be given.

Reply: the SEM morphology of fracture surfaces of ABS has been added in the [Figure](#)

9.

Figure 9. SEM morphology of fracture surfaces of the ABS/AINP composites (a-a₁: ABS; b-b₁: ABS-AINP₂₀; c-c₁: ABS-AINP₂₅; d-d₁: ABS-AINP₃₀)

Appendix B

Dear Dr Laura Smith, Professor Anthony Stace and Dr Chaohua Cui,

Thank you and referee for your kindness and patience to our manuscript (Manuscript ID: RSOS-200800.R1; Title: Synthesis of a novel aluminum salt of nitrogen-containing alkylphosphinate with high char formation to flame retard acrylonitrile–butadiene–styrene). Our manuscript is approaching to the level of publication with your help.

We have corrected the errors, according to the advice of the Referees. The amendment was blighted in yellow. In addition, we have submitted the following items:

- (1) A text file of the manuscript
- (2) A separate electronic file of each figure (.TIF)
- (3) A 100-word media summary of paper. (Lay abstract)
- (4) The raw data (it has been provided before reviewing)

The reply to the referee's comments

Major comments:

1. For the flame retardancy of polymers, char usually is made of carbon, which is produced from the consumption of polymeric materials. However, in this manuscript, the residues after the decomposition of ABS/AINP composites are mainly from AINP, which includes a very limited amount of carbon. The authors also mentioned this point in the section of 3.2 Thermal analysis. Therefore, the protective layer formed by AINP is more like a ceramic-like structure, rather than a char layer. The authors should correct this

point in the manuscript.

Reply: We have corrected 'char layer' into 'residues' in the whole manuscript.

2. "The AINP contributed to flame inhibition mainly by compact phosphorus/aluminum-rich residues in condensed phase". For this statement in the Abstract, it is incorrect. Flame inhibition means the reaction in the gas phase.

Reply: This sentence has been corrected into 'flame retardancy' The compact phosphorus/aluminium-rich substance acted as a barrier to enhance flame-retardant properties of the ABS.

3. In Line 19 of Page 21, no Figure 8b2 was included in this manuscript. Please put the right figure number here.

Reply: Figure 8b2 should be Figure 8b1. We have corrected it.

4. There are still many typo errors in the manuscript. Just name a few of them here: "retadancy" in Line 30 of Page 12; "STM D3801" in Line 1 of Page 15...Please do careful proofreading and check the grammar of the whole manuscript before the submission of the manuscript.

Reply: We have checked the manuscript again and corrected the type and grammar errors